# Heat-Killed *Lactococcus lactis* subsp. *cremoris* H61 Altered the Iron Status of Young Women: A Randomized, Double-Blinded, Placebo-Controlled, Parallel-Group Comparative Study

**DOI:** 10.3390/nu14153144

**Published:** 2022-07-30

**Authors:** Mizuki Takaragawa, Keishoku Sakuraba, Yoshio Suzuki

**Affiliations:** Graduate School of Health and Sports Science, Juntendo University, Inzai 270-1695, Japan; mizuki854@gmail.com (M.T.); sakuraba@kf6.so-net.ne.jp (K.S.)

**Keywords:** ferritin, probiotics, serum iron, transferrin saturation, unsaturated iron-binding capacity

## Abstract

Women are prone to iron deficiency because of increased iron excretion associated with menstruation. This is often treated by oral iron supplementation, although this treatment can cause side effects, such as stomach pain and nausea, with low absorption of ingested iron. Previously, a significant increase in serum iron was observed in association with the consumption of foods containing *Lactococcus lactis* subsp. *cremoris* H61 (H61). However, the causal relationship between H61 ingestion and elevated serum iron is still unclear. Therefore, in this study, we aimed to determine the effects of H61 ingestion on the iron status of young women. Healthy young Japanese women (18–25 years of age) ingested either heat-killed H61 or placebo for 4 weeks. Serum iron, transferrin saturation, and ferritin were significantly elevated in the H61 group but remained unchanged in the placebo group. Compared to before the intervention, iron intake remained unchanged during the intervention period, so the change in the iron status of the H61 group was not due to increased iron intake. These results suggest that heat-killed H61 may elevate iron status by enhancing iron absorption.

## 1. Introduction

Iron is an important nutrient because it is involved in the electron transport system and oxygen transport in the human body. However, due to its low absorption from food [1,2], iron deficiency is among the most common nutritional problems worldwide [3]. Although there is no physiological pathway that actively excretes iron, approximately 1–2 mg/day of iron is inevitably lost due to epithelial sloughing, sweating, and bleeding [4,5]. In addition, menstruating women are prone to iron deficiency anemia due to high iron discharge associated with menstruation [6].

Iron is absorbed from food in the intestine and enters circulation. Then, the iron released from enterocytes into the circulatory system is inhibited by the hepatic hormone hepcidin [7,8,9]. As serum iron concentration is tightly regulated by hepcidin, its oversecretion causes anemia, whereas its undersecretion causes iron overload [7]. Exercise [10,11] and excessive iron intake [12] enhance hepcidin secretion, so physically active women are more susceptible to iron deficiency anemia, even with adequate iron intake.

Iron deficiency is generally treated with oral iron supplementation. Due to the rapid increase in iron exposure in the gastrointestinal tract, oral supplementation may cause side effects, such as stomach pain and nausea [13]. An increase in serum iron was observed in people consuming either yogurt fermented with *Lactococcus lactis* subsp. *Cremoris* H61 (H61) [14] or a commercial dietary supplement containing heat-killed H61 [15]. However, in previous studies, participants were not instructed to refrain from iron supplementation during the intervention period [14], and the dietary supplement used contained other probiotics and vitamins [15]. Therefore, the causal relationship between H61 ingestion and elevated serum iron is not yet clarified.

Thus, in this study, a double-blind, randomized, placebo-controlled trial was conducted to determine the effect of H61 on the iron status of healthy young women.

## 2. Materials and Methods

### 2.1. Participants

A total of 50 young women were recruited. The inclusion criteria were: (1) healthy women aged 18–25 years (2) in good physical condition (3) willing to voluntarily participated in the study and provide written consent. The exclusion criteria were as follows: (1) a history of serious cardiovascular, hepatic, renal, respiratory, endocrine, or metabolic disorders; (2) a history of chest pain or syncope; (3) those at risk of developing allergies related to test supplements; (4) those who had 200 mL of blood drawn within 1 month or 400 mL within 3 months prior to the start of this study, (e.g., blood donation); (5) smokers; and (6) those who were otherwise deemed unsuitable by the study investigator. The purpose and methods of this study were explained to participants orally and in writing, and written consent was obtained.

The participants were randomly allocated to the test (H61) or placebo group. One participant in the placebo group withdrew prior to the intervention. Another participant in the placebo group was excluded because her initial serum iron concentration was not normal. During the intervention, six participants in the H61 group dropped out because they did not appear in the second blood sampling (*n* = 5) and lost the test supplement (*n* = 1). Forty-two participants completed the intervention. Among them, 13 participants were excluded for common cold (*n* = 4), iron use (*n* = 6), compliance of <80% (*n* = 3), or missing data (*n* = 1). The remaining 29 participants were analyzed (Figure 1, Table 1).

This study was conducted according to the Declaration of Helsinki (approved in 1964, amended in 2013), with the approval of the ethics committee of Juntendo University Graduate School of Health and Sports Science (approve No. 29-162). The study was also registered in UMIN-CTR (UMIN000030815) prior to its initiation.

### 2.2. Experimental Design

A randomized, double-blinded, placebo-controlled, parallel-group comparative trial was conducted. The participants took H61 (60 mg/day) or placebo at the same time daily for 4 weeks. During the intervention, the participants were instructed not to change their dietary habits. Blood was drawn before (pre) and after (post) the intervention to assess iron status. A dietary survey was conducted using a brief-type, self-administered diet history questionnaire (BDHQ) within 3 days before and after blood sampling. The participants recorded their test supplement intake, physical condition, medications/dietary supplement use, and menstrual status in a daily logbook during the intervention.

### 2.3. Test Supplement

The H61 supplement contained heat-killed H61 (Appendix A). One tablet contained 30.0 mg of heat-killed H61, 167.5 mg dextrin, 50.0 mg crystalline cellulose, and 2.5 mg calcium stearate. The placebo tablet contained 197.5 mg dextrin, 50.0 mg crystalline cellulose, and 2.5 mg calcium stearate. The appearance of the H61 and placebo supplements was indistinguishable.

The test supplements were provided by Toa Biopharma (Tokyo, Japan). These were packaged in aluminum pouches for each individual identified by a unique key code. The key codes were kept by Toa Biopharma and opened after the intervention was completed and all data were fixed.

### 2.4. Blood Collection and Measurements

Blood was collected from the cubital vein between 12:00 and 13:00. Blood and biochemical analyses were performed in a certified clinical laboratory (SRL, Tokyo, Japan). Briefly, red blood cell count, hemoglobin (Hgb), and hematocrit were assessed using a Sysmex XE-2100 automated hematology analyzer (Sysmex Corporation, Hyogo, Japan). Serum ferritin, serum iron, and total iron-binding capacity (TIBC) were evaluated using latex agglutination turbidimetry, direct colorimetry, and 2-nitroso-5-(N-propyl-N-sulfopropylamino) phenol (nitroso-PSAP) methods, respectively, using a JCA-BM8060 automatic analyzer (JEOL Ltd., Tokyo, Japan). Transferrin saturation (TSAT) and unsaturated iron-binding capacity (UIBC) were calculated as serum iron/TIBC × 100 and TIBC−serum iron, respectively.

Serum hepcidin concentrations were measured in duplicate using a hepcidin-25 extraction-free ELISA (Cosmo Bio, Tokyo, Japan) according to the protocol provided by the manufacturer. If the difference was <10% of the mean, the mean value was used as the concentration; otherwise, the measurement was repeated until the difference was <10%.

### 2.5. Dietary Survey

A dietary survey was conducted using BDHQ, a self-administered questionnaire developed for the Japanese population that was previously validated [16]. BDHQ was used to measure nutrient intake for one month from the time of the survey. The daily intake of iron and vitamin C was estimated according to the density of each nutrient (mg/1000 kcal) and estimated energy requirements (2200 kcal/day). The estimated energy requirement was based on physical activity level category III (high) for women aged 18–29 as defined in the Dietary Reference Intakes for Japanese (2015 edition), as all participants were active and belonged to a collegiate athletic club (e.g., basketball and soccer).

### 2.6. Menstrual Cycle

The menstrual cycle on the day of blood collection was classified into two phases according to the diary kept by participant: follicular or luteal [17]. The luteal phase was defined as up to 14 days from the first day of bleeding, whereas the follicular phase was defined as any other day.

### 2.7. Statistical Analyses

The changes in hematological parameters were analyzed using a generalized estimated equation of a generalized linear model controlled for menstrual cycle. The model included a subject ID as a subject variable, intervention (H61, placebo), measure point (Pre, Post), and menstrual cycle as within-subject variables and interactions; intervention × measure point was fixed, whereas intervention × menstrual cycle and measure point × menstrual cycle were included in the model if they reduced the quasi-information criterion. The model with the smallest criterion was adopted.

Statistical significance was set at *p* < 0.05. Statistical analyses were conducted using SPSS Statistics ver. 24 (IBM Japan, Tokyo, Japan).

## 3. Results

### 3.1. Iron Status

After 4 weeks of intervention, serum iron (*p* < 0.05), TSAT (*p* < 0.05), and ferritin (*p* < 0.001) increased significantly, whereas UIBC decreased significantly in the H61 group (*p* < 0.001). In the placebo group, MHC (*p* < 0.001) and hepcidin (*p* < 0.01) levels decreased significantly. A significant decrease in MCV was also observed in both groups (*p* < 0.05).

After the intervention (Post), the H61 group had significantly higher serum iron (*p* < 0.05), TSAT (*p* < 0.05), and ferritin (*p* < 0.05) and significantly lower UIBC (*p* < 0.01) than the placebo group. However, before the intervention (Pre), the H61 group also had significantly higher serum iron (*p* < 0.05) and TSAT (*p* < 0.01) and significantly lower UIBC (*p* < 0.01) than the placebo group (Figure 2, Appendix A).

Iron status, depending on menstrual cycle (follicular phase vs. luteal phase), was also examined. Ferritin was significantly higher (*p* < 0.05) in the follicular phase (EMM 33.9, SE 1.0 ng/mL) than in the luteal phase (EMM, 28.5; SE, 1.2 ng/mL). There were no significant differences in other parameters, such as serum iron and TSAT (Appendix A).

Throughout the study, the highest ferritin and TSAT were 103.0 ng/mL and 47.8%, respectively; therefore, no subjects were suspected of having iron overload.

### 3.2. Iron and Vitamin C Intake

During the intervention, the participants’ iron and vitamin C intake did not change relative to pre-intervention. The mean iron intake was lower than the estimated average requirement (EAR) in both groups, with 64% (9/14) of the H61 group and 67% (10/15) of the placebo group below the EAR during the intervention. The mean vitamin C intake was greater than the EAR in both groups, with 29% (4/14) of the H61 group and 40% (6/15) of the placebo group below the EAR during the intervention (Table 2).

Table 2 summarizes the intergroup differences. The estimated daily intake of iron and vitamin C did not differ significantly between the H61 and placebo groups pre- and post-intervention.

## 4. Discussion

The 4 weeks of heat-killed H61 consumption significantly increased serum iron, ferritin, and TSAT levels and significantly decreased UIBC in healthy women. Iron and vitamin C were not supplemented during the intervention. The estimated daily iron and vitamin C intake during the intervention did not differ from pre-intervention levels.

TSAT was calculated as Fe/TIBC, whereas UIBC was calculated as TIBC−Fe. Because serum iron increased, whereas TIBC did not change in the H61 group, the increase in serum iron may account for the increase in TSAT and decrease in UIBC.

When multiple measurements are taken, if a population with extreme initial values is selected, the next measurement will be closer to the overall mean. This is a statistical phenomenon called regression toward the mean, and examples in clinical parameters include blood pressure and serum cholesterol [18]. In this study, the participants were randomly allocated before the intervention, and pre-intervention serum iron was significantly higher in the H61 group (EMM, 103.8 µg/dL; SE, 4.4 µg/dL) than in the placebo group (EMM, 75.2 µg/dL; SE, 11.8 µg/dL). According to the National Health and Nutrition Survey, the mean serum iron level of Japanese women aged 20−29 years is 75.2 µg/dL (SD, 39.6 µg/dL) [19], suggesting that participants with high serum iron levels were disproportionately allocated to the H61 group. Therefore, if the intervention had no effect on serum iron, the post-intervention serum iron in the H61 group should have been lower than the pre-intervention level because of regression toward the mean. However, after the intervention, serum iron (EMM, 140.8 µg/dL; SE, 17.1 µg/dL) was significantly elevated in the H61 group, whereas it was unchanged in the placebo group. Thus, the four weeks of administration of heat-killed H61 may have increased the serum iron levels of healthy young women, even with a higher serum iron population.

Previously, an increase in serum iron was observed in young women consuming 300 mL/day of yogurt fermented with H61 for 4 weeks [14]. However, in this study, the mean serum iron was also elevated in the control group. Furthermore, a dietary survey was not included, and iron supplement use could not be excluded because the participants were female track and field athletes. Thus, the change in serum iron levels was not discussed as the influence of H61. In another study, serum iron increased when men took a commercially available dietary supplement containing H61 (1.6 × 10^8^ cells before being heat-killed/day) for 30 days [15]. However, the dietary supplement also contained another probiotic, *Lactobacillus sporegenes*, and vitamins (vitamin C, vitamin E, niacin, calcium pantothenate, vitamin B1, vitamin B6, vitamin B2, vitamin A, folic acid, vitamin D, and vitamin B12). Therefore, H61 could not be concluded as responsible for the increase in serum iron. In this double-blind, randomized, placebo-controlled study, the effect of heat-killed H61 was examined, and an increase in serum iron was observed.

The National Health and Nutrition Survey reported a mean daily iron intake of 7.1 mg (SD, 3.1 mg) for Japanese women aged 20–29 years and a mean vitamin C intake of 73 mg (SD, 52 mg) [19], which is involved in iron absorption [20]. Although a direct comparison of intakes cannot be made because of the difference in survey methods, the iron and vitamin C intakes of the participants in this study, although suboptimal, were not significantly different from those of average Japanese women aged 20–29 years. In addition, iron or vitamin C supplementation was not provided in this study, and the estimated dietary iron and vitamin C intakes during the intervention did not differ from pre-intervention levels. Thus, the increase in serum iron observed in the H61 group was not due to increased iron or vitamin C intake.

It is noteworthy that H61 increased the serum iron concentrations in healthy women with suboptimal dietary iron intake. Orally ingested iron is absorbed in the duodenum and upper small intestine. Heme iron is absorbed as-is, whereas non-heme iron (Fe^3+^) is reduced to Fe^2+^ by ferric reductase in the apical membrane [21] and is absorbed by divalent metal transporter 1 (DMT1) and human copper transporter 1 [22]. Therefore, the absorption of non-heme iron competes with absorption of zinc and copper, which are also absorbed by DMT1 [23]. Iron absorbed in enterocytes is converted to Fe^3+^ by transmembrane copper-dependent ferroxidase hephaestin and is released into the blood by the basal membrane iron transporter ferroportin (SLC40A1) and binds to transferrin in the serum [7,20]. The hepatic hormone hepcidin inhibits iron transport by ferroportin. [24]. In this study, serum iron was significantly increased in the H61 group. Therefore, H61 may have an influence at some point in this process.

Dietary iron absorption is estimated to be 14% with a Swedish diet, 16% with a French diet, and 16.6% with a U.S. diet [25]. Dietary non-heme iron (Fe^3+^) must be reduced to divalent iron (Fe^2+^) in the lumen before it can be absorbed. Vitamin C promotes iron absorption because it can reduce iron [20,26]. Lactate also enhances iron absorption [26,27]. The ingestion of heat-killed H61 was reported to increase the abundance of intestinal *Lactobacillales* [28]. *Lactobacillales*, commonly called lactic acid bacteria, ferment carbohydrates to produce lactate. Therefore, the ingestion of heat-killed H61 may have increased the abundance of *Lactobacillales* to increase the concentration of lactate in the intestines, enhancing iron absorption. However, the intestinal microbiota was not examined in this study. As the composition of the microbiota is differs considerably between humans and animals, the changes in the intestinal microbiota in rats resulting from heat-killed H61 ingestion cannot be directly applied to humans. Meanwhile, a recent systematic review showed that *Lactobacillus plantarum* 299v increases iron absorption [29]. Thereby, it seems possible that H16 may also increase iron absorption. Therefore, the mechanism to increase serum iron should be clarified.

Serum ferritin reflects the amount of stored iron [30]. In this study, the H61 group had an increased serum ferritin and serum iron after the intervention. Therefore, stored iron also seemed to increase in this group.

Iron status fluctuates with the menstrual cycle. Serum iron, TSAT, and hepcidin decrease before menstruation; recover from the onset to the end of menstrual bleeding; and peak during the follicular phase [31]. In female cyclists, Hgb and serum ferritin levels increase from the menstrual phase to the follicular phase and decrease from the follicular phase to the luteal phase [32]. In this study, serum ferritin was higher in the follicular phase than in the luteal phase, similar to a previous report [32], although TSAT and Hgb did not differ according to menstrual cycle. Thus, the menstrual cycle should be incorporated when assessing a woman’s iron status.

In a previous report, menstrual cycle was not considered, although an increase in serum iron was observed in young women who consumed yogurt fermented with H61 [14]. In this study, generalized estimating equations were used to control for menstrual cycle. Therefore, the results of this study may demonstrate the effects of heat-killed H61 without the influence of the menstrual cycle.

The treatment of iron deficiency anemia typically involves oral iron supplementation. However, such treatment has adverse effects, such as stomach pain and nausea, which can lead to poor compliance [13]. In contrast, the increase in serum iron observed in this study was not due to increased iron intake. None of the participants complained of gastrointestinal symptoms during the intervention. Therefore, instead of oral iron supplementation, H61 can be used to improve iron deficiency anemia without causing gastrointestinal symptoms or physical discomfort.

This study is subject to some limitations. We did not determine the amount of menstrual bleeding and individual differences in this parameter. This may have impacted the participants’ iron status. The initial serum iron level significantly differed between the participants in the H61 and placebo groups; this was an accidental bias resulting from random allocation upon entry. However, the possible difference in the participants’ iron bioavailability or genetic predispositions cannot be ruled as having influenced the results. Therefore, the effects of H61 should be re-examined in a population with uniform iron metabolism by measuring the amount of iron loss due to menstrual bleeding. In this study, the 4-week consumption of heat-killed H61 improved iron status with no signs of iron overload. However, it is necessary to determine the risk of iron overload with larger doses and/or longer duration.

## 5. Conclusions

Four-week consumption of heat-killed *Lactococcus lactis* subsp. *cremoris* H61 was demonstrated to increase serum iron, TSAT, and ferritin levels and decrease UIBC. As the improvement in iron status was not attributed to the amount of iron intake, dietary iron availability was suggested to be enhanced by heat-killed H61.

## Figures and Tables

**Figure 1 nutrients-14-03144-f001:**
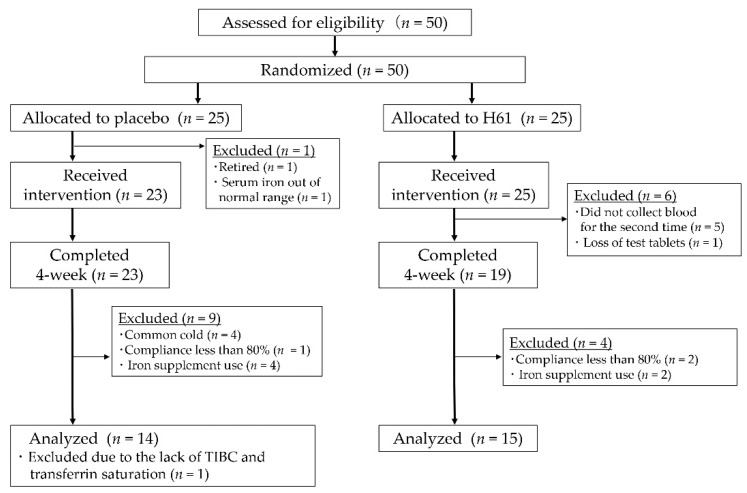
Flow diagram of the participants.

**Figure 2 nutrients-14-03144-f002:**
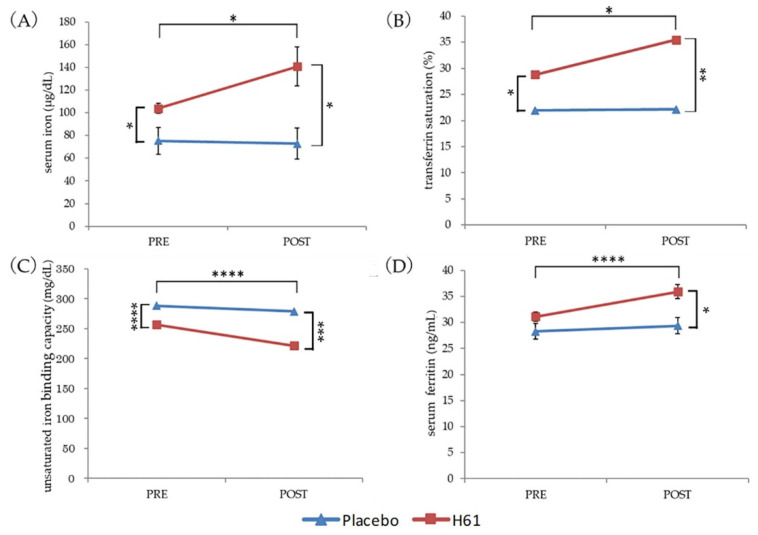
Changes in hematological parameters. (**A**) Serum iron. (**B**) Transferrin saturation. (**C**) Unsaturated iron-binding capacity. (**D**) Serum ferritin. * *p* < 0.05, ** *p* < 0.01, *** *p* < 0.005, **** *p* < 0.001.

**Table 1 nutrients-14-03144-t001:** Characteristics of the participants.

	Placebo (*n* = 15)	H61 (*n* = 14)	*p*
Age (years)	19.9 ± 1.5	19.9 ± 1.1	0.772
Height (cm)	163.0 ± 6.4	161.7 ± 6.7	0.603
Body weight (kg)	58.5 ± 5.9	56.6 ± 5.5	0.381

Data are expressed as means ± SD.

**Table 2 nutrients-14-03144-t002:** Changes in the estimated daily intake of iron and vitamin C.

	EAR	Group	PRE			POST			*p* ^b^
			Mean	SD	*p* ^a^	Mean	SD	*p* ^a^	
Iron (mg/day)	8.5 ^#^	H61	8.2	1.6	0.215	7.8	2.4	0.766	0.361
		placebo	7.5	1.3		7.6	1.9		0.900
Vitamin C (mg/day)	85	H61	126.5	44.4	0.234	110.8	51.8	0.779	0.121
		placebo	107.6	38.5		105.9	40.6		0.891

EAR, estimated average requirement; ^#^, women with menstruation aged 18–29; *p*
^a^, H61 vs. placebo; *p* ^b^, PRE vs. POST.

## Data Availability

The data presented in this study are available upon request from the corresponding author. The data are not publicly available due to ethical restrictions.

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
