# Peer review of "Heat-Killed Lactococcus lactis subsp. cremoris H61 Altered the Iron Status of Young Women: A Randomized, Double-Blinded, Placebo-Controlled, Parallel-Group Comparative Study"

_nutrients, 2022, doi:10.3390/nu14153144_

Round 1

Reviewer 1 Report

The manuscript reports relevant findings, linking the supplementation with heat-killed Lactococcus lactis H61 with increased body iron levels via (likely) increased dietary absorption. The study seems to be adequately designed, carefully performed and well documented. However, I fully agree with its limitations, especially I am concerned about the differences between the two subgroups at the baseline (maybe indeed the data could be biased by some genetic predispositions between the groups). But likely these limitations are not sufficient to disprove the study.

I would have 3 comments/suggestions:

- I am not an expert in the statistical methods that the authors used. This could be double-checked. I was surprised to see the larger statistical significance for the changes that look rather mild compared with the others (eg, Fig 2D).

- The Discussion part is in my opinion rather too long for the amount of data that this manuscript reports.

- More explanation could be provided on how heat-killed bacteria may affect iron absorption. I would expect this rather from alive organisms.

Author Response

Our response is attached.

Reviewer 2 Report

In the paper, the authors address an important nutritional problem for which a good way to counteract it still does not exist, despite the fact that much work has been devoted to it and various solutions have been proposed. They present an interesting, alternative and, as they conclude, effective method of improving iron status by increasing the availability of iron in the diet.

The plan and method of conducting the supplementation experiment are presented in detail and clearly described. The choice of parameters for evaluating the effects of supplementation is also comprehensive and allows for a comprehensive picture of changes in blood parameters. Despite these strengths, the manuscript there is the lack of possibility to interpret the results.  My objections are related to the supplement's little information.

A)  Why did the authors use heat-killed Lactococcus lactis subsp. cremoris H61? What was the extent of cell destruction? Is there any study of the composition of the supplement?  This could be very interesting due to recent reports in the literature of its positive effects on the skin, on the hair and now also on the body's iron status.  Shouldn't they have been done?

 B)  Based on previous studies, the authors suggest that the observed result is due to the prebiotic effect of the tested supplement. Referring to a study conducted on rats and the changes in the microbiota that it induced there, I believe is an insufficient argument for concluding that the observed effect is the result of the same changes.   

C)  A statistically significant increase in serum iron was observed was in a group in which this parameter was already suboptimal. The results of this study show that it would be necessary to check whether there would be further accumulation of Fe to levels that would be dangerous to health. 

Notes on the manuscript.  The "Conslusion" section in its current form is not well-developed. It is more of a summary.

General concern: Was the research sponsored in any way by Toa Biopharma Co.?

In conclusion, undoubtedly, the manuscript presents an interesting and promising work, however, the manuscript presents insufficient research.  The explanation of the results based on previous studies is insufficient.  

Author Response

Our response is attached.

Reviewer 3 Report

I have read an interesting paper by Takaragawa et al., entitled “Heat-killed Lactococcus lactis subsp. cremoris H61 altered the iron status of young women: A randomized, double blinded, placebo-controlled, parallel-group comparative study”. I think that the design of this study is properly carried out and presented.

I want to ask the author if they consider that the amount of menstrual bleeding (quantity or in days) is different between participant and if it may have an impact on iron deficiency? If you find it proper, please add some information in the discussion section, at least as a limitation.

I have a few minor comments:

- line 39 – “tis side effects” – please correct

- lines 38-46 – “However, this 38 rapidly increases….”, “Previously, an increase”, “However, in previous studies” – please try to avoid repeating words in the same paragraph

- line 53 – please delete “in this study”

- lines 51-61 – “50 healthy young women aged 18–25 years were recruited. The inclusion criteria were: 1) healthy women aged 18––25 years” – why repeating information?

- lines 51-61 – maybe you can present the inclusion and exclusion criteria more punctual

- lines 210-212 – “Although a direct ….., although suboptimal” – please try to rewrite

Author Response

Our response is attached.
